

# Minimally invasive pyeloplasty versus open pyeloplasty for ureteropelvic junction obstruction in infants: a systematic review and meta-analysis

Min Wang[1,*], Yu Xi[2,*], Nanxiang Huang[1], Pengli Wang[1], Li Zhang[1], Mingjia Zhao[1] and Siyi Pu[1]

[1] Nanchong Central Hospital, The Second Clinical College, North Sichuan Medical College, Nanchong, China
[2] Nanchong Hospital of Traditional Chinese Medicine, Nanchong, China
[*] These authors contributed equally to this work.

Corresponding author
Min Wang, 819203381@qq.com

## ABSTRACT

**Background**. To compare the perioperative outcomes and success rates of minimally invasive pyeloplasty (MIP), including laparoscopic and robotic-assisted laparoscopic pyeloplasty, with open pyeloplasty (OP) in infants.

**Materials and Methods**. In September 2022, a systematic search of PubMed, EMBASE, and the Cochrane Library databases was undertaken. The systematic review and meta-analysis were conducted in accordance with PRISMA (Preferred Reporting Items for Systematic Reviews and Meta-Analyses) guidelines, with the study registered prospectively in the PROSPERO database (CRD42022359475).

**Results**. Eleven studies were included. Dichotomous and continuous variables were presented as odds ratios (OR) and standard mean differences (SMD), respectively, with their 95% confidence intervals (CI). Compared to OP, a longer operation time and shorter length of stay were associated with MIP (SMD: 0.96,95% CI: 0.30 to 1.62, $p = 0.004$, and SMD: $-1.12$, 95% CI: $-1.82$ to $-0.43$, $p = 0.002$, respectively). No significant differences were found between the MIP and OP in terms of overall postoperative complications (OR:0.84, 95% CI: 0.52 to 1.35, $p = 0.47$), minor complications (OR: 0.76, 95% CI: 0.40 to 1.42, $p = 0.39$), or major complications (OR: 1.10, 95% CI: 0.49 to 2.50, $p = 0.81$). In addition, a lower stent placement rate was related to MIP (OR: 0.09, 95% CI: 0.02 to 0.47, $p = 0.004$). There was no statistical difference for success rate between the MIP and OP (OR: 1.35, 95% CI: 0.59 to 3.07, $p = 0.47$). Finally, the results of subgroup analysis were consistent with the above.

**Conclusions**. Our meta-analysis demonstrates that MIP is a feasible and safe alternative to OP for infants, presenting comparable perioperative outcomes and similar success rates, albeit requiring longer operation times. However, it is essential to consider the limitations of our study, including the inclusion of studies with small sample sizes and the combination of both prospective and retrospective research designs.

## INTRODUCTION

Ureteropelvic junction obstruction (UPJO) is a prevalent etiology for hydronephrosis in children, contributing to 85–90% of cases. Presently, the occurrence of neonatal hydronephrosis oscillates between 0.6% and 5.4%, as determined by current prenatal ultrasound diagnostics and urological assessments (*Chertin et al., 2006*). The conventional Anderson-Hynes pyeloplasty, since its first documentation in 1949 (*Anderson & Hynes, 1949*) has served as the gold standard for managing UPJO. Although open pyeloplasty (OP) boasts a success rate exceeding 90%, its prolonged recovery duration and suboptimal cosmetic outcomes diminish its appeal (*Chertin et al., 2006*; *O'Reilly et al., 2001*). In contrast, minimally invasive pyeloplasty (MIP), promising shorter postoperative recovery periods, diminished pain management requirements, superior cosmetic results, and a success rate comparable to OP, has garnered significant interest in recent years. This includes laparoscopic pyeloplasty (LP) and robot-assisted laparoscopic pyeloplasty (RALP) (*Chang et al., 2015*; *Cundy et al., 2014*).

Despite numerous reports of successful MIP in children, this technique remains underexplored in infants due to restricted abdominal space and challenges associated with internal suturing. However, with advancements in minimally invasive surgical techniques, *Kutikov, Resnick & Casale (2006)* were the first to report the safety and efficacy of LP in infants. Subsequent studies have reinforced the successful application of both RALP and LP in infants (*Andolfi et al., 2021*; *Zamfir Snykers et al., 2019*). Concurrently, comparative studies have shown MIP to be equally effective in infants and older children (*Chandrasekharam et al., 2021*; *Kawal et al., 2018*). While ample evidence supports MIP's superiority over OP in pediatric cases, there is a paucity of systematic evidence supporting MIP use in infants. Therefore, we undertook a systematic review and meta-analysis to assess MIP's efficacy in infants in comparison to OP.

## METHODS AND METHODS

This meta-analysis was conducted and reported in accordance with the PRISMA (Preferred Reporting Items for Systematic Reviews and Meta-Analyses) guidelines and prospectively registered in the PROSPERO database (CRD42022359475) (*Page et al., 2021*).

### Literature search

In September 2022, a systematic and exhaustive literature search was undertaken in PubMed, EMBASE, the Cochrane Library, CNKI, Wangfang and VIP databases. Search terms included "pyeloplasty", "laparoscopic", "robot", and "infant" (Table S1 for detailed search formula). The search was confined to publications in English or Chinese from 2000 onwards. Additionally, necessary references were manually retrieved from the PubMed database.

The inclusion criteria were as follows: (1) Studies focusing on infants with UPJO undergoing MIP; (2) studies that compared one or more of the surgical outcomes between MIP and OP, including operation time, length of stay (LOS), success rate, stent placement rate, and postoperative complications. The exclusion criteria included: (1) Studies not

specifically examining infants with UPJO; (2) studies in which participants did not undergo either MIP or OP; (3) research not comparing the specified surgical outcomes between MIP and OP; (4) letters, case reports, reviews, and conference abstracts; (5) studies with incomplete data or lacking relevance to the topic.

## Quality assessment

The quality of the included studies was evaluated using the Newcastle–Ottawa Scale (NOS), with a score of ≥7 classified as high quality (*Au et al., 2001*). The risk of bias in the included studies was assessed using The Risk of Bias in Non-Randomized Studies–of Interventions tool (ROBINS-I) (*Sterne et al., 2016*).

## Data extraction

Data extracted included author, publication year, study design, number of patients, participant characteristics (age, weight, intervention, male: female ratio, and follow-up), operation time, LOS, success rate, stent placement rate, and postoperative complications. All complications were recorded, defined, and graded according to the Clavien system (*Dindo, Demartines & Clavien, 2004*).

The above steps (literature search, quality assessment, data extraction) were independently carried out by two of us (M.W and Y.X). All discrepancies were resolved by the senior author (NX.H) after open discussions.

## Statistical analysis

Statistical analysis was performed using Review Manager software (RevMan) Version 5.3. Dichotomous and continuous variables were expressed as odds ratios (OR) and standard mean differences (SMD), along with their 95% confidence intervals (CI), respectively. A fixed-effects model was adopted when the $I^2$ value was less than 50%, indicating low to moderate heterogeneity, whereas a random-effects model was utilized when the $I^2$ value exceeded this threshold, signaling potential moderate to high heterogeneity, as suggested by *Higgins & Thompson (2002)*. It's worth noting that the $I^2$ statistic quantifies the proportion of the total variation in study estimates that is due to heterogeneity rather than chance. In light of potential concerns regarding the arbitrariness of the 50% cut-off, we have also conducted and provided sensitivity analyses using a fixed-effects model even for instances where $I^2$ exceeded 50%. Subgroup analysis was performed according to the type of MIP. A *p*-value of less than 0.05 was considered statistically significant.

## RESULTS

Following the removal of duplicates, irrelevant, and low-quality studies, a total of 11 studies (*Andolfi et al., 2022*; *Bansal et al., 2014*; *Cui et al., 2022*; *Dangle et al., 2013*; *García-Aparicio et al., 2014*; *Kafka et al., 2019*; *Kallas-Chemaly et al., 2019*; *Masieri et al., 2019*; *Rague et al., 2022*; *Tanaka et al., 2008*; *Tong et al., 2009*) were included (Fig. 1). The detailed characteristics of the included studies, along with their quality evaluations, are presented in Table 1. Four studies reported on RALP, six on LP, with one study reporting on both. According to the ROBINS-I tool, two studies were identified as high risk, while the rest were deemed moderate risk (Fig. 2).
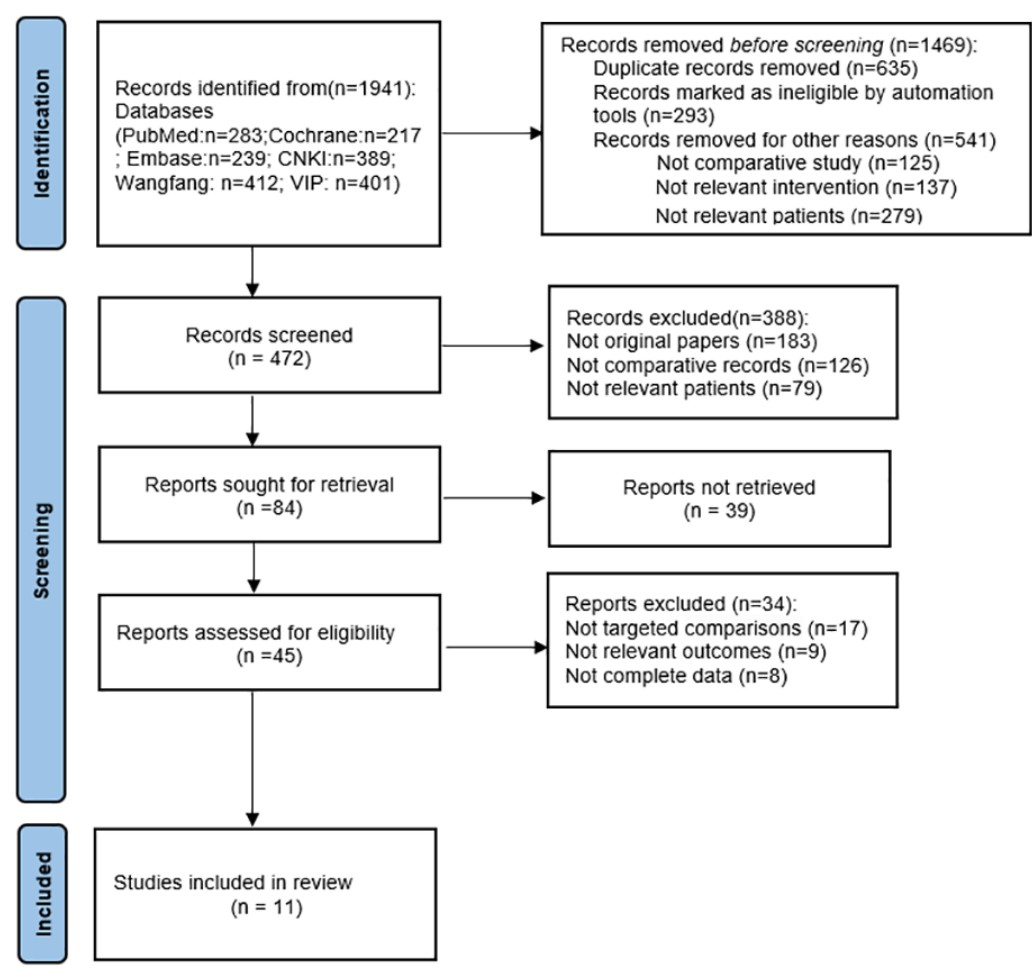

**Figure 1 PRISMA flow diagram.**

## Perioperative outcomes

MIP was associated with a longer operation time and a shorter LOS (SMD: 0.96, 95% CI: 0.30 to 1.62, $p = 0.004$, and SMD: $-1.12$, 95% CI: $-1.82$ to $-0.43$, $p = 0.002$, respectively, Fig. 3). Subgroup analysis indicated no significant difference in operation time between RALP and OP (SMD: 0.32, 95% CI: $-0.46$ to 1.10, $p = 0.42$, Fig. S1), whereas the LP group had a longer operation time than OP (SMD: 1.46, 95% CI: 0.52 to 2.40, $p = 0.002$, Fig. S1). Furthermore, no statistical significance was found in LOS for either RALP or LP compared with OP (SMD: $-0.86$, 95% CI: $-1.93$ to 0.22, $p = 0.12$, and SMD: $-0.95$, 95% CI: $-2.03$ to 0.13, $p = 0.08$, respectively, Fig. S2).

## Complications

The overall postoperative complications, minor postoperative complications (Clavien grade I-II), and major postoperative complications (Clavien grade III-V) were analyzed respectively. No statistical difference was found between MIP and OP in terms of overall postoperative complications, minor complications, and major complications (OR: 0.84,

Wang et al. (2023), *PeerJ*, DOI 10.7717/peerj.16468

**Table 1  Characteristics and quality assessment of included studies.**

| Author,year | Study Design | Intervention | No.of patients | Age (months) | Weight (kg) | M:F | Follow-up (months) | Quality Score[f] |
|---|---|---|---|---|---|---|---|---|
| *Andolfi et al. (2022)* | R | RALP VS LP VS OP | 39/26/39 | 4(2-6)/3(2-7)/7(4-7)[a] | NA | 30:9/22:4/28:11 | 15(8-26)/20(14-43)/10(4-23)[a] | 8 |
| *Bansal et al. (2014)* | R | RALP VS OP | 9/61 | 9.2(3.7-11.9)/4(1-11.6)[b] | 8(5.8-10.9)/7(4-14)[b] | 4:5/47:14 | 10(7.2-17.8)/43.6(3.4-73.8)[b] | 8 |
| *Cui et al. (2022)* | R | LP VS OP | 32/34 | $7.2 \pm 2.3/6.7 \pm 2.8$[c] | $8.1 \pm 2.9/7.6 \pm 3.7$[c] | 22:10/23:11 | NA | 7 |
| *Dangle et al. (2013)* | R | PALP VS OP | 10/10 | 7.3(2-12)/3.31(1-10)[d] | 8.01(5.1-13.1)/7.9(6.1-12.8)[d] | 8:2/9:1 | 9.01(4-24)/18.1(5.7–23.8) | 8 |
| *García-Aparicio et al. (2014)* | R | LP VS OP | 26/32 | $5.15 \pm 2.98/4.25 \pm 3.06$[c] | $7.02 \pm 1.96/6.78 \pm 1.94$ | 22:4/24:8 | NA | 7 |
| *Kafka et al. (2019)* | P | RALP VS OP | 15/15 | 7(3.5-11)/7(4-12)[b] | 7(5.6-9.8)/6.3(4.8-10)[b] | NA | NA | 8 |
| *Kallas-Chemaly et al. (2019)* | R | LP VS OP | 24/53 | $7.1 \pm 3.87/5.2 \pm 2.6$[c] | $7.97 \pm 1.7/7.67 \pm 1.3$ | 19:5/41:12 | NA | 8 |
| *Masieri et al. (2019)* | R | LP VS OP | 9/9 | 10(7-12)/6.3(4-9)[d] | 9.14(7.4-10)/8.01(7-9.3) | 5:4/6:3 | 26/27.1[e] | 8 |
| *Rague et al. (2022)* | R | RALP VS OP | 83/121 | 7.2(5.9-9.4)/2.9(1.9-5)[a] | 8.2(7.2-9.3)/5.9(5-9)[a] | 65:18/87:34 | 16.1(6.5-37.1)/34.1(13-57.2)[a] | 8 |
| *Tanaka et al. (2008)* | R | LP VS OP | 38/2541 | $9 \pm 5.76/7.68 \pm 5.64$[c] | NA | 28:10/1856:695 | NA | 7 |
| *Tong et al. (2009)* | R | LP VS OP | 23/21 | 7.1(2-11)/8.2(3-12)[b] | NA | 13:10/12:9 | 19(6-36)/24(12-48)[b] | 8 |

**Notes.**

R, retrospective; P, prospective; RALP, robot-assisted laparoscopic pyeloplasty; VS, versus; LP, laparoscopic pyeloplasty; OP, open pyeloplasty; No., number; M:F, male:female; L:R, left:right; NA, not available.

[a]Median (interquartile interval)
[b]Median (range)
[c]Mean $\pm$ Standard Deviation, SD
[d]Mean(range)
[e]Mean
[f]Using NOS scoring Rules.

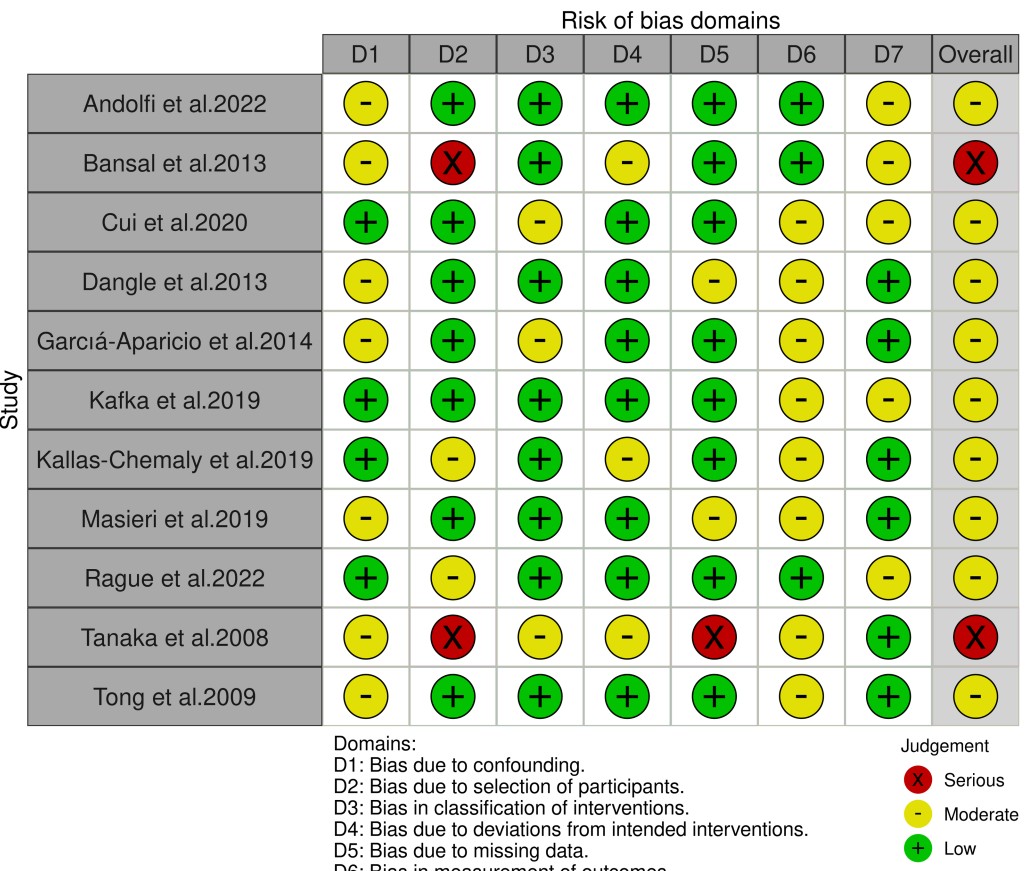

Figure 2 The risk of bias for the included studies according to ROBINS-I.

95% CI: 0.52 to 1.35, $p = 0.47$, OR: 0.76, 95% CI: 0.40 to 1.42, $p = 0.39$, and OR: 1.10, 95% CI: 0.49 to 2.50, $p = 0.81$, respectively, Fig. 4). Subgroup analysis results echoed these findings (Fig. S3).

## Stent placement rates

Seven studies included in the analysis reported postoperative stent placement. The combined analysis demonstrated that MIP was associated with a lower stent placement rate compared to OP (OR: 0.09, 95% CI: 0.02 to 0.47, $p = 0.004$, Fig. 5).

## Success rates

No statistical difference was observed in terms of the success rate between MIP and OP (OR: 1.35, 95% CI: 0.59 to 3.07, $p = 0.47$, Fig. 5). Subgroup analysis also found no statistical difference in the success rate between RALP and OP (OR: 1.75, 95% CI: 0.66 to 4.63, $p = 0.26$, Fig. S4) or LP and OP (OR: 0.68, 95% CI: 0.16 to 2.81, $p = 0.59$, Fig. S4).

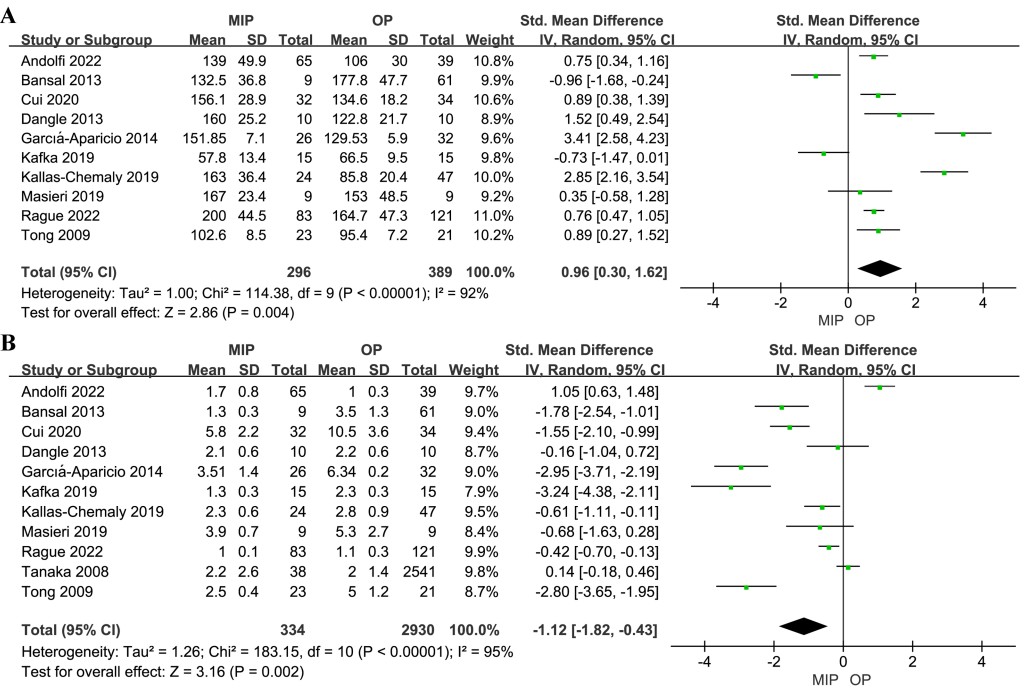

**Figure 3** Forest plot and meta-analysis of operation time (A) and LOS (B).

## Sensitivity analyses

To ensure the robustness of our findings against model selection, we executed sensitivity analyses using a fixed-effects model for outcomes with $I^2$ exceeding 50%. While the primary meta-analysis showed little variation between fixed and random-effects models, some subgroup analyses were notably influenced by the model choice, altering the statistical significance of outcomes. This underscores the significance of model choice in meta-analyses, especially in subgroups. These analyses are elaborated in Table S2. Considering the studies with small samples and mixed designs (prospective and retrospective), we conducted another sensitivity analysis for outcomes with high heterogeneity. Neither study size nor design notably influenced the outcomes for operation time and LOS. These details are in Table S2. Furthermore, after omitting two high-risk bias studies, our findings remained largely unchanged, with detailed results also available in Table S2.

## Publication bias

Funnel plots were utilized to assess potential publication bias for the outcomes. There was significant publication bias present in most of the findings, such as in the overall postoperative complications (Fig. 6), which demonstrated significant asymmetry.

## DISCUSSION

Since the inaugural report of pediatric LP in 1995 (*Peters, Schlussel & Retik, 1995*), MIP has garnered favor among pediatric surgeons due to its superior LOS, enhanced postoperative pain management, improved cosmetic results, and comparability in terms of success

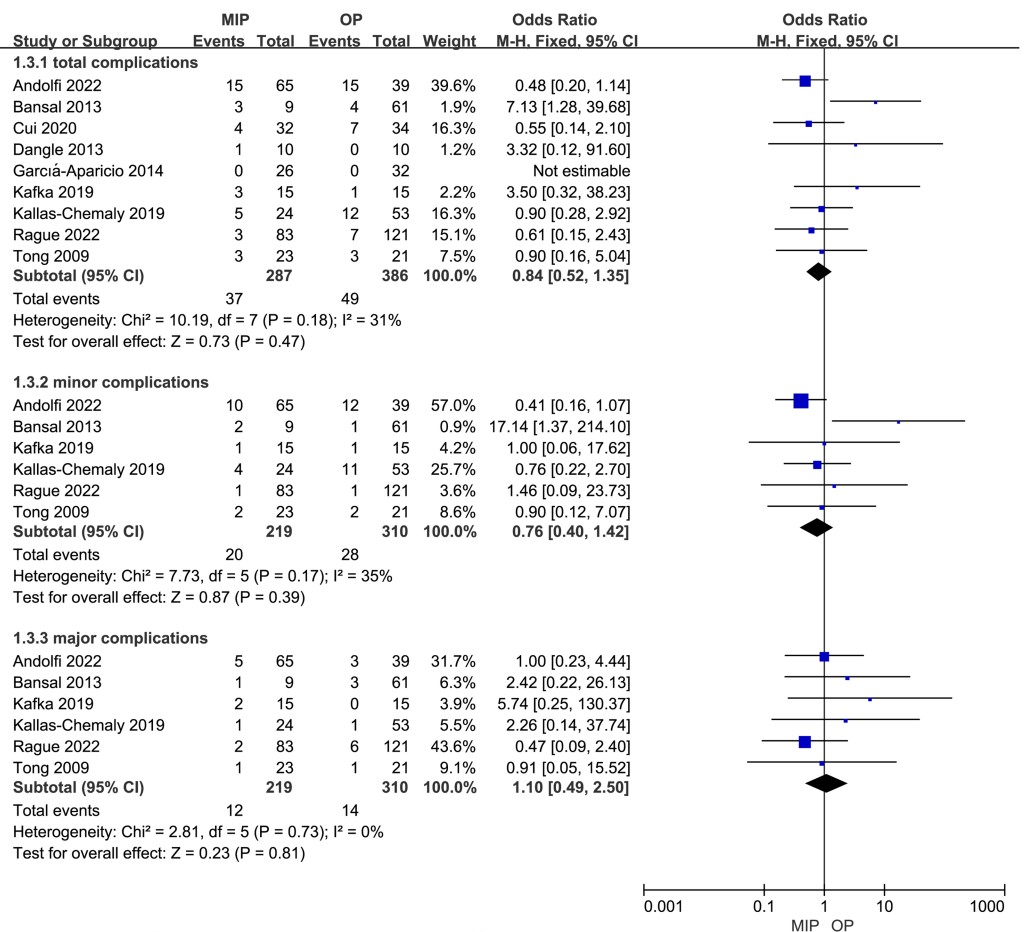

**Figure 4** Forest plot and meta-analysis of postoperative complications.

rates and complications to OP (*Andolfi et al., 2020*; *Gatti et al., 2017*). From 2003 to 2015, utilization rates of both OP and LP have been observed to decrease at an annual rate of 10% and 12% respectively, whereas RALP saw an annual increase of 29%. Notably, among RALP recipients, children (45%) and adolescents (20%) experienced the most pronounced increases (*Varda et al., 2018*). Nevertheless, the utilization rate of OP in infants has consistently exceeded 80%. This may be primarily attributed to the expedient recovery associated with OP, coupled with potential risks of hemodynamic and respiratory disorders affiliated with MIP (*García-Aparicio et al., 2014*; *Kallas-Chemaly et al., 2019*; *Varda et al., 2018*). As MIP methodologies mature and stabilize, numerous institutions have reported the procedure to be safe and feasible in infants, thereby implying that MIP is no longer exclusively dependent on patient age (*Baek et al., 2018*; *Denes et al., 2008*). Consequently, we undertook a systematic review to fortify the evidential basis for MIP use in infants.

The cumulative analysis demonstrated that OP had a shorter operation time than MIP. However, intriguingly, a subsequent subgroup analysis indicated that the operation time

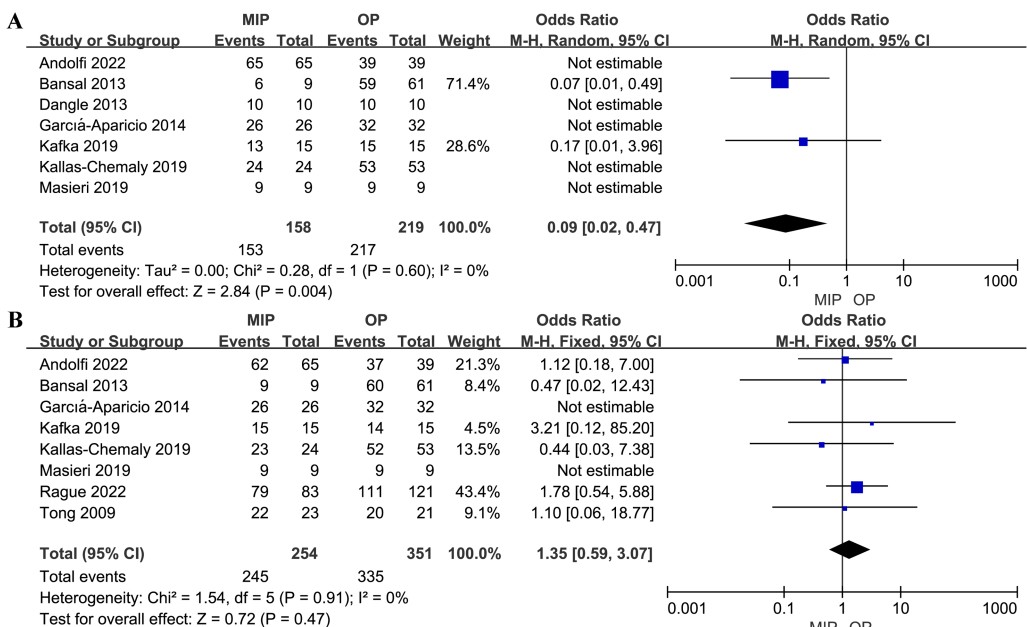

**Figure 5** Forest plot and meta-analysis of postoperative stent placement (A) and success rate (B).

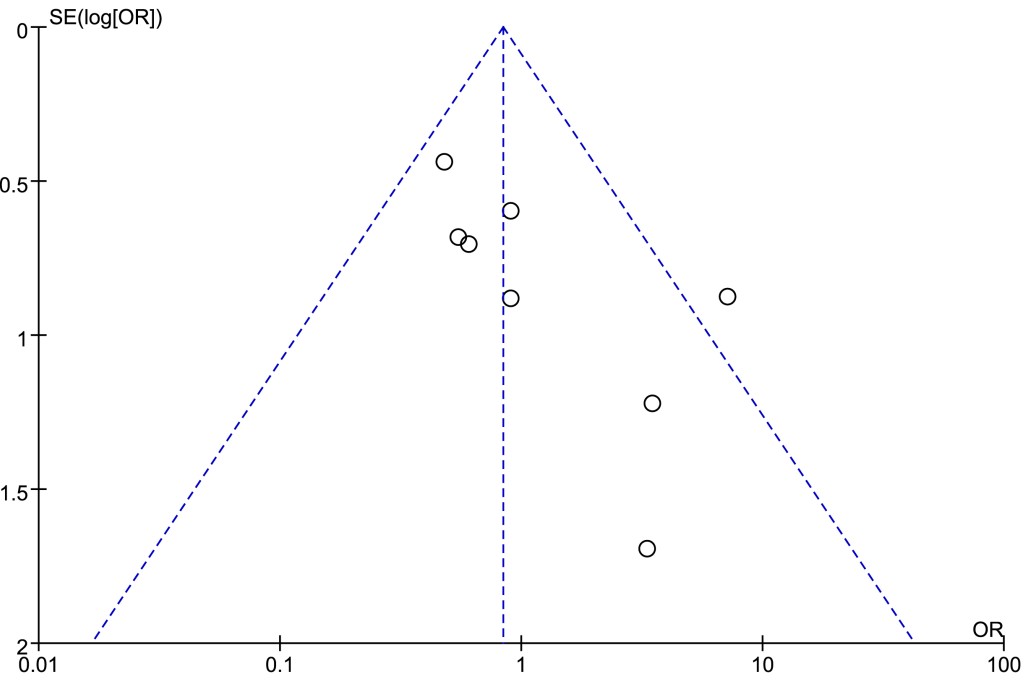

**Figure 6** Funnel plot of overall postoperative complications.

for OP was comparable to that of RALP, but shorter than that of LP. The robotic system, when compared to traditional LP, is ergonomic and affords a three-dimensional view, tremor filtering for motion scaling, and a wrist-like motion of the instruments, factors that may elucidate the observed differences in the subgroup analysis (*Andolfi et al., 2022*). The stringent criteria required for LP, which may induce mental stress, also contributed to these differences, particularly in infants (*Bansal et al., 2014*). The longer operation time of MIP when compared to OP was primarily associated with the complexities inherent to infant anatomy. The limited abdominal space and ureter size compounded the difficulty of correctly identifying the anatomical structure and executing suturing operations during MIP, thereby prolonging operation time. With regards to RALP, increased experience and proficiency significantly expedited the processes of robot installation and docking, which in turn correlated with the operation time (*O'Kelly, Farhat & Koyle, 2020*). In addition, the comprehensive analysis established a correlation between MIP and a shorter LOS. MIP wounds were found to have a higher collagen deposition postoperatively in comparison to OP, which facilitated quicker postoperative recovery and consequently reduced LOS (*Casale, 2012*; *Dangle et al., 2013*). The diminished requirements for tissue manipulation and analgesia with MIP also contributed to the shorter LOS (*Andolfi et al., 2022*). Furthermore, we discovered that a subset of OP practitioners preferred the utilization of Salle stents, which are typically removed prior to discharge, in contrast to the double-J tube. This preference resulted in an extended LOS (*García-Aparicio et al., 2014*; *Kallas-Chemaly et al., 2019*). Interestingly, a subgroup analysis discovered no significant differences in LOS between RALP or LP and OP. Prior to the stabilization of the learning curve, surgical confidence was typically lacking, prompting a more conservative approach to LOS, especially in the case of infants. This factor can provide a plausible explanation for the results of the subgroup analysis.

Regrettably, both OT and LOS presented significant heterogeneity, thereby compromising the quality of the results. Through subgroup and sensitivity analyses, we discerned that the high heterogeneity was primarily attributed to the learning curve, operator experience, and complex cases such as those involving crossing vessels, rather than the minimally invasive surgical approach itself. Additionally, variations in postoperative management across different institutions also contributed to the elevated heterogeneity of LOS, including disparate approaches to postoperative pain management and stent selection. *Andolfi et al. (2022)* reported that the learning curve for LP was steeper compared to that of RALP, with the first inflection point appearing in the 18th and 13th cases respectively. This suggests that achieving proficiency in LP necessitated a longer duration than RALP. Moreover, the learning curve for RALP reportedly stabilized after the 33rd case in infants, a finding that was congruent with trends observed in older children (*Andolfi et al., 2022*; *Dangle et al., 2013*; *Tasian, Wiebe & Casale, 2013*).

Regarding complications, our analysis yielded no statistical significance between MIP and OP in terms of overall postoperative complications, minor complications, or major complications, as observed in both the combined and subgroup analyses. The most frequently reported postoperative complications were stent-related, such as urinary tract infections and stent displacement (*Kafka et al., 2019*; *Rague et al., 2022*). *Rague et al. (2022)*

concluded through univariate and multivariate analysis that MIP did not constitute a risk factor for postoperative complications. Consequently, focus has been placed on ureteral stents as a means of mitigating postoperative complications. Beyond stent-related complications, the removal of ureteral stents also necessitates secondary anesthesia. Consequently, several institutions have pursued stent-free procedures following MIP and OP, with encouraging results (*Kocvara et al., 2014*; *Rodriguez, Rich & Swana, 2012*; *Silva et al., 2015*). However, stent-free procedures necessitated longer indwelling drainage, thus prolonging LOS and increasing costs (*Kafka et al., 2019*). Intriguingly, while our pooled analysis determined that the stent placement rate following MIP was lower than OP, no statistical significance was observed in postoperative complications between the two groups. This suggests that going stent-free does not necessarily circumvent complications. On the contrary, ureteral stents can decrease complications such as urinary leakage and urinoma (*Elmalik, Chowdhury & Capps, 2008*; *Smith et al., 2002*). Therefore, the question of whether to place ureteral stents post-pyeloplasty remains a subject of debate and warrants further investigation. Separately, several studies reported that the complication rates and success rates associated with externalized uretero-pyelostomy stents were comparable to those of standard ureteral stents, deeming them a safe alternative to traditional ureteral stents (*Braga et al., 2008*; *Lee et al., 2015*). Furthermore, a few studies suggested that continuous prophylactic antibiotics could reduce postoperative infection complications, though the quality of evidence was weak and not universally applicable across all pediatric patients (*Braga et al., 2013*; *Silay et al., 2017*).

Our pooled analysis revealed no significant difference in the success rate between MIP and OP. In their survival curve analysis, *Rague et al. (2022)* found that the median time for OP failure was 12.4 months, compared to 5.4 months for RALP. We surmise that complex cases significantly impact the success rate of MIP. In line with this perspective, *Lucas et al. (2012)* posited that prior endopyelotomy and crossing vessels considerably reduced the success rate of MIP. Additionally, *Tong et al. (2009)* suggested that the complications were linked with the failure of pyeloplasty, including fibrotic scarring induced by urine leakage, and ureter deformation caused by overtightening of sutures during surgery. While no conversions from MIP to OP were reported in the studies included in our analysis, surgeons have recommended several measures to minimize the conversion rate and enhance the success rate in infants, given their small abdominal space and heightened sensitivity to $CO_2$, intra-abdominal pressure, and hypothermia. To enlarge the abdominal space, *Masieri et al. (2019)* advised sufficient preoperative bowel preparation with simethicone and enemas prior to surgery, whereas *Andolfi et al. (2020)* suggested the placement of a nasogastric tube for decompression, utilizing an endorectal tube if necessary, without any preoperative bowel preparation (*Andolfi et al., 2020*; *Masieri et al., 2019*). To avoid hypothermia, the control of room temperature and the use of warmed $CO_2$ to establish pneumoperitoneum were suggested, although the use of heating blankets remains a subject of debate due to the potential risk of hyperthermia (*Andolfi et al., 2020*; *Zamfir Snykers et al., 2019*). Regarding intraperitoneal pressure, a randomized controlled trial found that maintaining an intraperitoneal pressure of 6–8 mmHg was optimal for infants and even conducive to early recovery (*Sureka et al., 2016*). Additionally, *Andolfi et al. (2020)* recommended

waiting at least 30 s after each instrument exchange to allow pneumoperitoneum to fully recover.

The cost associated with MIP continues to be a topic of considerable interest. *Casella et al. (2013)* reported that there was no statistically significant difference in cost between RALP and LP. Comparatively, *Dangle et al. (2013)* found no statistically significant difference in the direct cost of RALP and OP ($p = 0.10$). When considering the implications of parental productivity, *Behan et al. (2011)* discovered that each patient undergoing RALP resulted in average savings of $90.01 in parental wages lost and $612.80 in hospital expenses, even after the amortized cost of the robot was accounted for. Hence, the shorter LOS associated with MIP can offset its high cost. In conclusion, parents of children who underwent RALP reported significantly higher satisfaction levels with regards to ''overall life'', confidence, self-esteem, the burden of postoperative follow-up, and the size of the incision scar as compared to the OP group (*Barbosa et al., 2013*; *Freilich et al., 2010*).

Our analysis has several inherent limitations that warrant discussion. Firstly, all studies included in our analysis were observational. Although these types of studies provide valuable real-world data, they come with an inherent risk of bias. The varied effect sizes observed in our Forest Plot results potentially hint at methodological differences or clinical variations among the incorporated studies. Secondly, even though our sensitivity analysis indicated that including two studies with small sample sizes (less than 30 participants) and mixing both prospective and retrospective study designs did not drastically influence the heterogeneity or pooled results, such inclusions can still diminish statistical power. This may result in either missing significant findings or inadvertently introducing biases. Lastly, there is the undeniable concern of publication bias. Studies with significant findings are often more likely to be published, while those with null or negative results might be overlooked or remain unpublished. This bias can skew the overall effect size and may overestimate the benefits or underestimate the harms. To counteract these limitations, we employed sensitivity analyses and subgroup analyses, and emphasized both qualitative and quantitative insights. We recommend interpreting our findings with these limitations in mind and suggest future work to incorporate a broader range of studies, including unpublished data.

While our study offers insights into the efficacy of MIP compared to OP in infants, it is rooted in specific clinical settings and infant populations. Outcomes may vary due to surgeon expertise, regional healthcare infrastructure, and evolving techniques. Thus, extrapolating these findings to diverse contexts necessitates prudence.

## CONCLUSIONS

In conclusion, our meta-analysis indicates that MIP serves as a viable and safe alternative to OP in infants, demonstrating comparable perioperative outcomes and similar success rates, despite the extended operation time associated with MIP. Nonetheless, due to the quality of evidence currently available, further high-quality clinical studies are necessitated to thoroughly evaluate the efficacy of MIP in the infant population.

### Funding

The authors received no funding for this work.

### Competing Interests

The authors declare there are no competing interests.

### Author Contributions

- Min Wang conceived and designed the experiments, performed the experiments, authored or reviewed drafts of the article, and approved the final draft.
- Yu Xi conceived and designed the experiments, performed the experiments, authored or reviewed drafts of the article, and approved the final draft.
- Nanxiang Huang conceived and designed the experiments, authored or reviewed drafts of the article, and approved the final draft.
- Pengli Wang analyzed the data, prepared figures and/or tables, and approved the final draft.
- Li Zhang performed the experiments, analyzed the data, prepared figures and/or tables, and approved the final draft.
- Mingjia Zhao performed the experiments, analyzed the data, prepared figures and/or tables, and approved the final draft.
- Siyi Pu analyzed the data, prepared figures and/or tables, and approved the final draft.

### Data Availability

The raw data is available in the Supplemental File.

### Supplemental Information

Supplemental information for this article can be found online at http://dx.doi.org/10.7717/peerj.16468#supplemental-information.

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
