# Peer review of "Minimally invasive pyeloplasty versus open pyeloplasty for ureteropelvic junction obstruction in infants: a systematic review and meta-analysis"

_PeerJ, doi:10.7717/peerj.16468_

## Round 0.1 · original submission · Major Revisions

Please address all issues cautiously.

Reviewer 1 ·

Basic reporting

I would suggest the authors to spell "MIP" on the abstract (Laparoscopic and Robotic assisted laparoscopic").
This will make easier to the reader, when screening the abstract, understand which approach is being consider under the label of MIP.

We know that the article was done in China and there is a tendency to include our own language on the review. However, the authors should comment on what was the reason for this inclusion, and if it could have changed the results?

Experimental design

No comment

Validity of the findings

I think the discussion section is quite thorough, however the limitations paragraph is short.
I would suggest elaborating a bit more on the limitations. For example, expand a bit more on the Forest Plot results, publication bias and possible implications on the results of this MA.

Additional comments

This a well-written article. The English language is clear and easy to follow.
The design, collection of data and analysis follow the accepted guidelines and clearly displayed on tables, charts and figures.
The conclusions are within the hypothesis set by the authors.
I would expand on the "limitations" paragraph.

Reviewer 2 ·

Basic reporting

This study relied on a SLR and meta-analysis to compare MIP and OP in infants. The methods and results are well-documented but I have some recommendations (as below).

Abstract: It would be nice to spell out the full name of some acronyms, e.g., SMD, OR, to be more friendly for readers with relatively limited stats background.

Line 24: “and major complications” --- Did you mean “or major complications”?

Line 29: If at all possible, it would be nice to add key limitations to Conclusion?

Line 87: Recommend adding reference to justify the cutoff of 50% for I-squared? Also recommend adding the interpretation of I-squared? I know what you meant but it would be clearer for non-technical readers. Also recommend reporting fixed-effect results (even if it’s just sensitivity analyses) even when I-squared is greater than 50% because some may argue that the use of 50% might be arbitrary.

Line 97: Recommend being very specific and precise as to which studies were excluded and why. For the two studies with high risk, what measures were taken? Excluding the two studies from the analyses in the base case or sensitivity analyses?

Results section: I don’t see sensitivity analyses reported. Is there any?

Line 241: Recommend elaborating more on the limitations. Right now, it’s very concise. For example, what do you mean by “insufficient data”, “all included observational studies”? Readers could benefit from more accurate details (e.g., some actual numbers in addition to qualitative statement). It would be nice to also elaborate a bit more what measures have been taken to mitigate these limitations and what future researchers could do.

Discussion: Recommend adding discussion about the generalizability of the study results? E.g., in what settings would the study conclusion stand and in what scenarios would the conclusion not stand, and why?

Experimental design

Standard study design. No additional comments other than those related to reporting (mentioned above)

Validity of the findings

Standard study design. No additional comments other than those related to reporting (mentioned above)

Additional comments

Standard study design. No additional comments other than those related to reporting (mentioned above)

Reviewer 3 ·

Basic reporting

This is well-written manuscript with a comprehensive literature search and meta-analysis for an interesting clinical question. I only have minor comments.

In the literature search section, the authors did not clearly specify the study type and study outcomes of interest to be included in the search and the following meta-analysis. Different study outcomes within different types of studies have different clinical implications. It’s important to clearly define these before conducting the search. Please clarify and elaborate.

Some of the studies the authors included in the meta-analysis only include a relatively small number of patients (e.g., less than 30 patients). Is the evidence summarized in the manuscript sufficient to support the conclusion?

One of the 11 studies is a prospective study vs other 10 retrospective studies. Is it appropriate to conduct the meta-analysis with all the 11 studies together? Please discuss.

Experimental design

No comments

Validity of the findings

No comments

---

## Round 0.2 · accepted · Accept

Thank you for addressing the concerns and suggestions of the reviewers.

Reviewer 1 ·

Basic reporting

no comment

Experimental design

no comment

Validity of the findings

no comment

Reviewer 2 ·

Basic reporting

Reviewers’ comments have been addressed in a clear and organized way. No further comment.

Experimental design

Reviewers’ comments have been addressed in a clear and organized way. No further comment.

Validity of the findings

Reviewers’ comments have been addressed in a clear and organized way. No further comment.